# Fair Graph Distillation

**Qizhang Feng[1], Zhimeng Jiang[1], Ruiquan Li[2], Yicheng Wang[1], Na Zou[1], Jiang Bian[3], Xia Hu[4]**
[1]Texas A&M University, [2]University of Science and Technology of China,
[3]University of Florida, [3]Rice University

## Abstract

As graph neural networks (GNNs) struggle with large-scale graphs due to high computational demands, graph data distillation promises to alleviate this issue by distilling a large real graph into a smaller distilled graph while maintaining comparable prediction performance for GNNs trained on both graphs. However, we observe that GNNs trained on distilled graphs may exhibit more severe group fairness issues than GNNs trained on real graphs for vanilla and fair GNNs training. Motivated by these observations, we propose *fair graph distillation* (FGD), an advanced graph distillation approach to generate fair distilled graphs. The challenge lies in the deficiency of sensitive attributes for nodes in the distilled graph, making most debiasing methods (e.g., regularization and adversarial debiasing) intractable for distilled graphs. We develop a simple yet effective bias metric, named coherence, for distilled graphs. Based on the proposed coherence metric, we introduce a framework for fair graph distillation using a bi-level optimization algorithm. Extensive experiments demonstrate that the proposed algorithm can achieve better prediction performance-fairness trade-offs across various datasets and GNN architectures.

## 1 Introduction

Real-world data, like chemical molecules, social networks, and transportation networks, can be represented as graphs [Han et al., 2022a, Ling et al., 2023a, Jiang et al., 2022a, Ying et al., 2018, Ling et al., 2023b, Tong et al., 2020, Han et al., 2022b]. Graph neural networks (GNNs) excel at capturing structural information but struggle with large-scale graphs due to memory consumption and computational expense caused by the neighborhood explosion problem[Hamilton et al., 2017, Liu et al., 2023c]. This cost becomes unaffordable in situations requiring repeated GNN training, such as neural architecture search and continual learning [Liu et al., 2018, Zhou et al., 2019, Li and Hoiem, 2017, Liu et al., 2023b]. Dataset distillation is a promising solution to address computation challenges by generating small, informative distilled data for neural network training in downstream tasks [Jin et al., 2021, 2022, Zhao et al., 2021a,b, Nguyen et al., 2021]. Techniques like dataset condensation [Zhao et al., 2021b, Jin et al., 2021] can significantly reduce training data size without major performance degradation in the image and graph domains. However, focusing solely on prediction performance may introduce fairness issues, as sensitive information can be condensed into distilled data for prediction. A natural question is raised: *Is the model trained on the distilled graph fair, and if not, how can we achieve fair graph distillation?*

In this work, we focus on the group fairness[1] for node classification tasks under binary sensitive attribute setting. We discover that GNNs trained on distilled small graphs exhibit more severe group fairness issues than those on real graphs. In other words, graph distillation can even *amplify* graph

---

[1]Group fairness ensures equitable treatment of diverse demographic groups by algorithms Mehrabi et al. [2021]. Such as in mortality prediction, issues arise when true positive rates significantly differ between sensitive groups. Group fairness metrics will be introduced in Section 5.1.

37th Conference on Neural Information Processing Systems (NeurIPS 2023).

data bias, which challenges the applicability of graph distillation in high-stake applications [Mehrabi et al., 2021, Suresh and Guttag, 2019]. To this end, we propose a fair graph distillation framework to offer a significantly reduced graph size and also better utility-fairness trade-off while maintaining predictive performance.

Many debias methods explicitly use sensitive attributes, but these are inherently missing in distilled graphs because they are excluded from the data attributes and the meaning of the attributes may change during the optimization process. In this paper, we point out the relationship between the space of real graphs and the space of distilled graphs and develop a simple estimator of sensitive attributes and introduce a bias measurement called consistency. We then propose a bi-level optimization algorithm for fair graph distillation: the outer loop generates a fair and informative distilled graph using gradient matching and coherence loss, while GNNs train on distilled graphs in the inner loop. In a nutshell, the contributions can be summarized as follows:

- To our knowledge, this is the first paper to identify group fairness issues in conventional graph distillation methods with binary sensitive attributes, motivating the formulation of a fair graph distillation problem in node classification tasks.
- We discover the relationship between the space of real graphs and the space of distilled graphs. We develop a bias metric called coherence for distilled graphs and propose a bi-level optimization framework using this metric to achieve fair graph distillation.
- We perform extensive experiments on various real-world datasets and GNN architectures to validate the effectiveness of the proposed FGD algorithm. Results demonstrate that FGD achieves a better accuracy-fairness trade-off compared to vanilla graph distillation methods and numerous baselines.

## 2 Preliminaries

### 2.1 Notations

We consider node classification tasks given a graph dataset $\mathcal{G} = \{\boldsymbol{A}, \boldsymbol{X}, \boldsymbol{Y}, \boldsymbol{S}\}$ with $N$ nodes. Here, $\boldsymbol{A} \in \{0,1\}^{N \times N}$ is the adjacency matrix, and $A_{ij} = 1$ represents there exists an edge between node $i$ and $j$. $\boldsymbol{X} \in \mathbb{R}^{N \times D}$ is the node feature matrix, where $D$ is non-sensitive feature dimension for each node. $\boldsymbol{Y} \in \{0,1,\cdots,C-1\}^N$ denotes the node labels over $C$ classes. For simplicity, we consider a binary sensitive attribute[2] $\boldsymbol{S} \in \{0,1\}^N$. $\boldsymbol{\Pi}^s$ is the sensitive membership diagonal matrix. $\boldsymbol{\Pi}^s_{ii} = 1$ if and only if $i$-th node belongs to sensitive group $s$. The distilled small graph dataset is marked as $\mathcal{G}' = \{\boldsymbol{A}', \boldsymbol{X}', \boldsymbol{Y}'\}$ which contains $N'$ nodes and $N' \ll N$. Note that elements of the distilled adjacency matrix satisfy $\boldsymbol{A}'_{ij} \in [0,1]$ and no sensitive attributes exist in $\mathcal{G}'$. The latent node representation of real graph is $\boldsymbol{Z}$, and the span space of it is $\text{span}(\boldsymbol{Z}) \coloneqq \left\{ \sum_{i=1}^N w_i \boldsymbol{z}_i | 1 \leq i \leq N, w_i \in \mathbf{R} \right\}$. Similar definition of $\boldsymbol{Z}'$ and $\text{span}(\boldsymbol{Z}')$ for distilled graph.

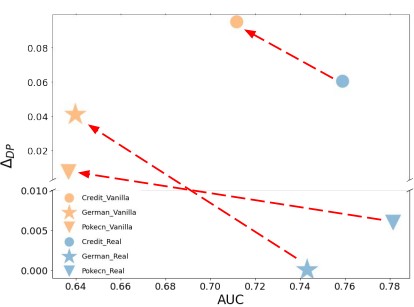

Figure 1: AUC and $\Delta_{DP}$ of the GNN trained on real graph data and distilled graph data. Both utility and fairness performance deteriorates after vanilla graph distillation.

### 2.2 Graph Distillation via Gradient Matching

The purpose of graph distillation is to generate a *distilled* graph $\mathcal{G}'$ such that the GNN model, denoted as $\text{GNN}_\theta$ with parameters $\theta$, trained on distilled graph performs *comparably* to the model trained on the real graph $\mathcal{G}$. The objective can be formulated as the following bi-level optimization problem:

$$\min_{\mathcal{G}'} \mathcal{L}\left(\text{GNN}_{\theta^{\mathcal{G}'}}\left(\boldsymbol{A}, \boldsymbol{X}\right), \boldsymbol{Y}\right) \quad \text{s.t} \quad \theta^{\mathcal{G}'} = \arg\min_{\theta} \mathcal{L}\left(\text{GNN}_\theta\left(\boldsymbol{A}', \boldsymbol{X}'\right), \boldsymbol{Y}'\right) \tag{1}$$

where $\theta^{\mathcal{G}'}$ denotes the optimal parameter trained on distilled small graph $\mathcal{G}'$, and $\mathcal{L}(\cdot, \cdot)$ denotes the loss function. To tackle the above optimization problem, the gradient matching method Zhao et al.

---

[2]The sensitive attribute $S$ represents the attribute that the respondents do not want to be disclosed, such as gender or age. Sensitive attribute $S$ is not included in the normal features $X$.

[2021a] is proposed. The intuition is to let the GNN parameters $\theta^{\mathcal{G}'}$ trained on distilled graph follow a similar path to the GNN parameters $\theta^{\mathcal{G}}$ trained on the real graph during model optimization. The gradient of the GNN parameters is forced to be the same over the real and distilled graphs:

$$\min_{\mathcal{G}'} \left[ \sum_{t=0}^{T-1} D\left(\nabla_\theta \mathcal{L}(\mathcal{G}), \nabla_\theta \mathcal{L}(\mathcal{G}')\right) \right], \tag{2}$$

where $D(\cdot, \cdot)$ is a distance function, $T$ is the number of steps of model parameters trajectory, and $\theta_t^{\mathcal{G}}, \theta_t^{\mathcal{G}'}$ denotes the model parameters trained on $\mathcal{G}$ and $\mathcal{G}'$ at time step $t$, respectively. The gradient calculated on $\mathcal{G}$ and $\mathcal{G}'$ is denoted as $\nabla_\theta \mathcal{L}(\mathcal{G}) \coloneqq \nabla_\theta \mathcal{L}\left(\text{GNN}_{\theta_t}\left(\boldsymbol{A}, \boldsymbol{X}\right), \boldsymbol{Y}\right)$ and $\nabla_\theta \mathcal{L}(\mathcal{G}') \coloneqq \nabla_\theta \mathcal{L}\left(\text{GNN}_{\theta_t}\left(\boldsymbol{A}', \boldsymbol{X}'\right), \boldsymbol{Y}'\right)$, respectively.

# 3    Bias Measurement for Distilled Graph

In this section, we empirically demonstrate the fairness issue in the distilled graph. Motivated by this, we pursue fair graph distillation. Although distilled graphs lack sensitive attributes $\boldsymbol{S}'$, we observe that between the node representations of the real and distilled graphs: their barycenter remain consistent, and their spaces are also consistent. We leverage this phenomenon to develop a simple, effective bias measurement for distilled graphs.

## 3.1    Is Graph Distillation Really Fair?

Our empirical investigation assesses the fairness of graph distillation across various datasets and architectures. We compare the utility (AUC) and fairness (demographic parity ($\Delta_{DP}$) [Beutel et al., 2017]) of GNNs trained on real graphs and those trained on distilled graphs created by the vanilla graph distillation method. The utility and fairness performance are shown in Figure 1. We can find that: For datasets like Pokec-n, German, and Credit, distilled graph-based GNNs have higher $\Delta_{DP}$ and lower AUC performance, suggesting a compromise in fairness and prediction performance. We also notice that, for Pokec-z and Recidivism datasets, these GNNs exhibit lower $\Delta_{DP}$ and significantly lower AUC performance (shown in Table 1), indicating a trade-off between improved fairness and reduced prediction performance. We observe similar results when using other fair GNN models. More details can be found in Appendix E. Motivated by these observations, we aim to find a better prediction performance and fairness trade-off via chasing the fair graph distillation method.

## 3.2    Geometric Connections in Data Distillation

The distilled data can be generated via minimizing gradient distance in Equation 2. To simplify the analysis, we consider $D(\cdot, \cdot)$ as Euclidean distance and those model parameters during optimization trajectory satisfying $\theta \sim \mathcal{P}_\theta$, where $\mathcal{P}_\theta$ is certain but unknown parameters' distribution. Therefore, the objective can be transformed as

$$\min_{\mathcal{G}'} \mathbb{E}_{\theta \sim \mathcal{P}_\theta} \left[ ||\nabla_\theta \mathcal{L}(\mathcal{G}) - \nabla_\theta \mathcal{L}(\mathcal{G}')||^2 \right]. \tag{3}$$

We consider three assumptions of the model parameters' distribution and the convergence for loss minimization.

**Assumption 3.1** (Model parameters' distribution). We assume that each model parameter in the last softmax layer satisfies the same distribution.

**Assumption 3.2** (Loss minimization). We assume that exists at least one distilled dataset that minimizes Equation 3.

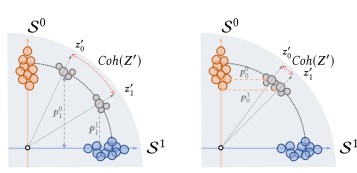

Figure 2: Geometric intuition of sensitive attribute estimation. The projection distance indicates the extent to which the node belongs to the sensitive group. (a) Unfair node representations have a large coherence bias. (b) Fair node representations have a small coherence bias.

**Theorem 3.3** (Consistent Span Space). *We empirically show that $\text{span}(\boldsymbol{Z}') \approx \text{span}(\boldsymbol{Z})$ via calculating the principle angles between them. We also provides the rigorous proof of $\boldsymbol{z}' \in \text{span}(\boldsymbol{Z})$ under distribution matching in Appendix D.*

**Theorem 3.4** (Consistent Geometric Barycenters). *Under Assumptions 3.1 and 3.2, the barycenter of the last representation for the optimal distilled graph and the real graphs are consistent, i.e. $\frac{1}{N} \sum_{i=1}^{N} \boldsymbol{z}_i = \frac{1}{N'} \sum_{i=1}^{N'} \boldsymbol{z}_i'$. Please see proof in Appendix B.*

### 3.3 Sensitive Attribute Estimation

The consistent span space and geometric barycenter suggest that we can estimate sensitive attributes from the representations of both distilled and real graphs. We frame sensitive attribute estimation as a classification problem: Given a data representation $\boldsymbol{z}' \in \boldsymbol{Z}'$, what is the probability that $\boldsymbol{z}'$ belongs to the sensitive group?

**Ridge regression for distance measurement.** Notice that the representation of each sensitive group for the real graph is known, we define $\boldsymbol{Z}_0$ and $\boldsymbol{Z}_1$ as the representation matrix for sensitive group $s = 0$ and $s = 1$. To measure the probability that $\boldsymbol{z}'$ belongs to these two sensitive groups, we first find the closest vector $\boldsymbol{z}'_{proj} = \boldsymbol{Z}_s^\top \boldsymbol{q} \in \text{Span}(Z_s)$ to approximate the representation $\boldsymbol{z}'$, and then use the norm of $\boldsymbol{z}' - \boldsymbol{z}'_{proj}$ to measure the distance between $\boldsymbol{z}'$ and sensitive group $\boldsymbol{Z}_s$. Specifically, we adopt ridge regression to find the optimal coefficient vector $\boldsymbol{q}*$, which can be formulated as

$$Dist(\boldsymbol{z}', \boldsymbol{Z}_s) = \|\boldsymbol{z}' - \boldsymbol{Z}_s^\top \boldsymbol{q}^*\|_2 \tag{4}$$

$$\text{s.t } \boldsymbol{q}^* = \arg\min_{\boldsymbol{q}} \gamma \|\boldsymbol{z}' - \boldsymbol{Z}_s^\top \boldsymbol{q}\|_2^2 + \|\boldsymbol{q}\|_2^2, \tag{5}$$

where $\gamma$ is the hyperparameter for ridge regression. For the optimal $q^*$, we have

$$\boldsymbol{p}^s = \boldsymbol{z}' - \boldsymbol{Z}_s^\top \boldsymbol{q}^* = \boldsymbol{z}' - \gamma \boldsymbol{Z}_s^\top (\boldsymbol{I} + \gamma \boldsymbol{Z}_s \boldsymbol{Z}_s^\top)^{-1} \boldsymbol{Z}_s \boldsymbol{z}', \tag{6}$$

where $\top$ represents matrix transpose, $\boldsymbol{p}^s$ is approximately the projection onto the orthogonal complement of the subspace $\text{span}(\boldsymbol{Z}_s)$. The proof is in Appendix C.

**Sensitive attribute soft estimation.** Since $\boldsymbol{p}^s = \boldsymbol{z}' - \gamma \boldsymbol{Z}_s^\top (\boldsymbol{I} + \gamma \boldsymbol{Z}_s \boldsymbol{Z}_s^\top)^{-1} \boldsymbol{Z}_s \boldsymbol{z}'$ can be viewed as approximately the projection of $\boldsymbol{z}$ onto the orthogonal complement of sensitive group $\boldsymbol{Z}_s$, $\|\boldsymbol{p}^s\|_2$ is small if $\boldsymbol{z}$ is in sensitive group $\boldsymbol{Z}_s$ and large otherwise. The probability of the given data representation $\boldsymbol{z}$ belongs to the sensitive group $\boldsymbol{Z}_s$ can further be inferred via a softmax function:

$$\pi^s(\boldsymbol{z}') = \frac{\exp\left(-\lambda \|\boldsymbol{p}^s\|_2\right)}{\sum_{s=0}^1 \exp\left(-\lambda \|\boldsymbol{p}^s\|_2\right)} \in [0, 1], \tag{7}$$

where $\lambda$ is the temperature hyperparameter. The sensitive attribute probability of $\boldsymbol{z}'$ for distilled graph can be estimated as probability distribution $[\pi^{s=0}(\boldsymbol{z}'), \pi^{s=1}(\boldsymbol{z}')]$, where $\pi^{s=0}(\boldsymbol{z}') + \pi^{s=1}(\boldsymbol{z}') = 1$.

### 3.4 Bias Measurement

Given the estimated sensitive attribute probability for $\boldsymbol{z}'$ of each distilled node, how can we measure the bias for them? For a fair representation, we can not distinguish which representation is more likely to be a specific sensitive group. Therefore, we adopt a simple surrogate bias measurement, named coherence, the variance of the estimated sensitive group membership. Given the whole distilled data representation $\boldsymbol{Z}' = [\boldsymbol{z}_1..., \boldsymbol{z}_{N'}]^\top$, the bias can be defined as:

$$Coh^s(\boldsymbol{Z}') = \widehat{Var}\left(\boldsymbol{\pi}^s(\boldsymbol{Z}')\right) = \frac{1}{N'} \sum_{n=1}^{N'} \left(\boldsymbol{\pi}^s(\boldsymbol{z}'_n) - \frac{1}{N'} \sum_{n=1}^{N'} \boldsymbol{\pi}^s(\boldsymbol{z}'_n)\right)$$

Note that $Coh^{s=0}(\boldsymbol{Z}') = Coh^{s=1}(\boldsymbol{Z}')$, and we adopt abbreviation $Coh(\boldsymbol{Z}')$ [3].

**Geometric intuition.** The intuition of sensitive attribute estimation, as illustrated in Figure 2, can be grasped from a geometric standpoint. In a toy example with a two-dimensional data representation, $\boldsymbol{z}' \in \mathbb{R}^2$, we consider two demographic groups for a binary sensitive attribute. The subspace spanned by the data representations from these groups is denoted by $\mathcal{S}^0$ and $\mathcal{S}^1$. Data representations to be estimated are $\boldsymbol{z}'_0$ and $\boldsymbol{z}'_1$. $\boldsymbol{p}_0^0$, $\boldsymbol{p}_1^0$ and $\boldsymbol{p}_0^1$, $\boldsymbol{p}_1^1$ is the projection of $\boldsymbol{z}'_0$ and $\boldsymbol{z}'_1$ onto the orthogonal complement of $\mathcal{S}^0$ and $\mathcal{S}^1$. As for fair data representation, zero coherent encourages all representations aligned with a "line" so that all representations are with the same normalized similarity with sensitive groups. Figure 2 (a) shows the case in which the data representation is biased where $\boldsymbol{z}'_0$ and $\boldsymbol{z}'_1$ can be easily distinguished. Figure 2 (b) shows that fairer data representation as they are less separable.

---

[3]For multiple-value sensitive attribute, we can use average coherence $Coh(\boldsymbol{Z}')$ across all sensitive groups.

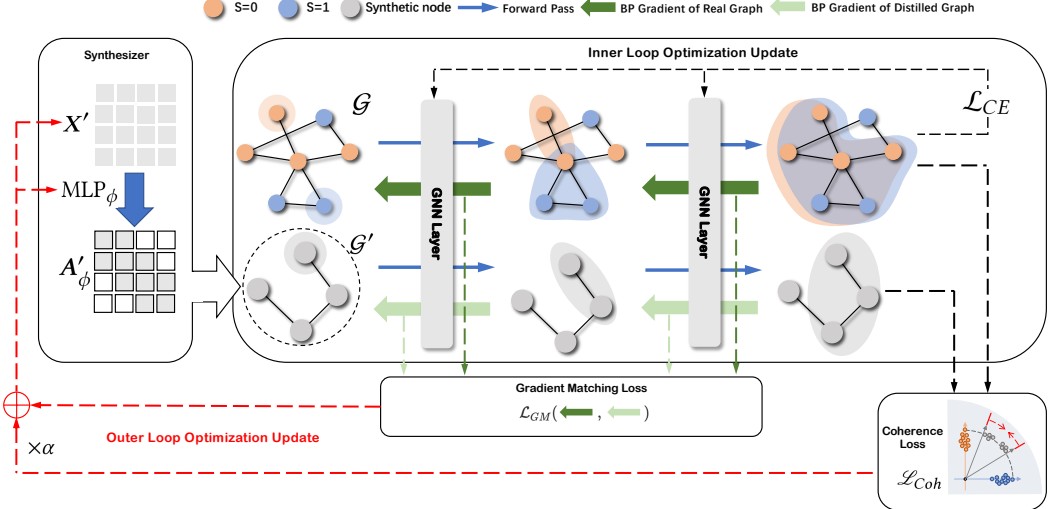

Figure 3: An overview of the proposed framework. The synthesizer generates the attribute matrix and adjacency matrix of the distilled small graph $\mathcal{G}'$. The Cross-Entropy loss $\mathscr{L}_{CE}$ guides the update of GNNs model during the inner optimization loop. Gradient matching loss $\mathscr{L}_{GM}$ and coherence loss $\mathscr{L}_{Coh}$ guide the update of the synthesizer during the outer optimization loop for utility and fairness.

## 4 Methodology

### 4.1 Problem Statement

Based on the proposed coherence metric, we argue that if $Coh(\mathbf{Z}')$ is reduced, bias in the distilled graph can be mitigated. As a result, if GNNs are trained on such distilled graphs, the bias issues in downstream tasks could also be alleviated. The problem is formally defined as: Given an undirected attributed network $\mathcal{G} = \{\mathbf{A}, \mathbf{X}, \mathbf{Y}, \mathbf{S}\}$, our goal is to obtain an debiased distilled graph $\mathcal{G}' = \{\mathbf{A}', \mathbf{X}', \mathbf{Y}'\}$ via reducing $Coh$, so that the fairness issues of GNNs trained on $\mathcal{G}'$ is mitigated. Hence the overall objective goal for generating a fair and condensed graph is:

$$\min_{\mathcal{G}'} \mathcal{L}_{GM} + \alpha \mathcal{L}_{Coh} \left( \text{GNN}_{\theta^{\mathcal{G}'}} \left( \mathbf{A}', \mathbf{X}' \right), \text{GNN}_{\theta^{\mathcal{G}'}} \left( \mathbf{A}, \mathbf{X} \right) \right)$$

$$\text{s.t } \theta^{\mathcal{G}'} = \arg\min_{\theta} \mathcal{L}_{CE} \left( \text{GNN}_{\theta} \left( \mathbf{A}', \mathbf{X}' \right), \mathbf{Y}' \right) \tag{8}$$

### 4.2 Fair Graph Distillation Loss

**Gradient Matching Loss.** We adopt gradient matching, as shown in equation (2), for graph distillation to distill useful information for node classification tasks. However, treating both $\mathbf{X}'$ and $\mathbf{A}'$ as learnable parameter [4] and directly optimizing $\mathbf{A}'$ is unaffordable due to $O(N^2)$ computation complexity. Following previous work Jin et al. [2021], we parameterize $\mathbf{A}'$ as a function of $\mathbf{X}'$:

$$\mathbf{A}'_{i,j} = \text{Sigmoid} \left( \frac{\text{MLP}_{\phi}([\mathbf{x}'_i; \mathbf{x}'_j]) + \text{MLP}_{\phi}([\mathbf{x}'_j; \mathbf{x}'_i])}{2} \right), \tag{9}$$

where $\mathbf{A}'_{i,j}$ is $i$-th row, $j$-th column of $\mathbf{A}'$, $\text{MLP}_{\phi}$ is a multi-layer neural network parameterized with $\phi$ and $[\cdot; \cdot]$ denotes concatenation. Note that $\mathbf{A}'$ is controlled to be symmetric since $\mathbf{A}'_{i,j} = \mathbf{A}'_{j,i}$. Sigmoid function pushes $\mathbf{A}'$ close to 0 or 1 to encourage its sparsity. For simplicity, we denote the parameterized adjacency matrix as $\mathbf{A}'_{\phi}$. In this way, we can reduce the complexity to $O(N)$.

The distance metric $D$ measures the similarity of gradients over the real graph and distilled graph. We adopt the summation of the gradient distance over all layers as the final gradient distance:

$$D \left( \nabla_{\theta} \mathcal{L}(\mathcal{G}), \nabla_{\theta} \mathcal{L}(\mathcal{G})' \right) = \sum_i \left( 1 - \frac{\nabla_{\theta} \mathcal{L}(\mathcal{G})_i \cdot \nabla_{\theta} \mathcal{L}(\mathcal{G})'_i}{\|\nabla_{\theta} \mathcal{L}(\mathcal{G})_i\| \|\nabla_{\theta} \mathcal{L}(\mathcal{G})'_i\|} \right) \tag{10}$$

---

[4] The distilled label $\mathbf{Y}'$ is sampled from real label $\mathbf{Y}$ with the same class probability, and it is fixed.

where $\nabla_\theta \mathcal{L}(\mathcal{G})_i$ is the i-th column vectors of the gradient matrices. Hence the loss objective for the graph distillation module is given by:

$$\mathscr{L}_{GM} = \mathbf{E}_{\theta \sim P_\theta} \left[ \sum_{t=0}^{T-1} D\left(\nabla_\theta \mathcal{L}(\mathcal{G}), \nabla_\theta \mathcal{L}(\mathcal{G}')\right) \right] \tag{11}$$

where $\mathcal{G}' = \{\boldsymbol{X}', \boldsymbol{A}', \boldsymbol{Y}'\}$, $t$ is the training epoch, and $\theta_t$ is well-trained GNNs model parameters. To reduce the impact of parameter initialization, the initial model parameters $\theta_0$ are sampled from a distribution of random initialization.

**Coherence loss.** In Section 3.4, we introduce coherence as a bias metric for distilled data. To mitigate bias, we use coherence bias as a regularization for fair synthesis. This calculation employs real graph node attribute $\boldsymbol{X}$ and distilled node attribute $\boldsymbol{X}'$ to estimate sensitive group memberships but overlooks structural bias in graph data. Given the GNN propagation mechanism, bias can exist in both node attributes and graph structure Dong et al. [2022]. Even without attribute bias, node representation may still be biased if structural bias is present.

Work by Dong et al. [2022] suggests that structural bias can be measured through graph representation bias. Leveraging this, we aim for low coherence in node attributes and representations to fully remove bias from our distilled graph. Specifically, we introduce attribute and structural coherence to decrease attribute and structural bias, respectively, by minimizing the variance in sensitive group membership estimation for node attributes and representations. Given a real graph data $\mathcal{G} = \{\boldsymbol{A}, \boldsymbol{X}, \boldsymbol{Y}, \boldsymbol{S}\}$ and a distilled graph $\mathcal{G}' = \{\boldsymbol{A}', \boldsymbol{X}', \boldsymbol{Y}'\}$, we feed them into a $L$-layer GNN, where the $l$-th layer latent representation in the GNN is denoted as $\boldsymbol{Z}_l$. The latent representation for node attribute after $l$-hop propagation contains both attribute bias as well as structural bias. Note the node attribute $\boldsymbol{X}$ and $\boldsymbol{X}'$ before propagation as $\boldsymbol{Z}_0$ and $\boldsymbol{Z}_0'$, we get a set of latent presentation $\{\boldsymbol{Z}_0, \boldsymbol{Z}_1, ..., \boldsymbol{Z}_L\}$ for $\mathcal{G}$ and $\{\boldsymbol{Z}_0', \boldsymbol{Z}_1', ..., \boldsymbol{Z}_L'\}$ for $\mathcal{G}'$. The objective to measure bias of $\boldsymbol{Z}_l'$ is:

$$Coh(\boldsymbol{Z}_l') = \widehat{Var}\left(\boldsymbol{\pi}^j(\boldsymbol{Z}_l')\right) = \widehat{Var}\left(\frac{\exp\left(-\lambda \|\boldsymbol{C}^j \boldsymbol{Z}_l'\|_2\right)}{\sum_j \exp\left(-\lambda \|\boldsymbol{C}^j \boldsymbol{Z}_l'\|_2\right)}\right), \tag{12}$$

where $\boldsymbol{C}^j = \gamma_j \left(\boldsymbol{I} + \gamma_j \boldsymbol{Z}_l \boldsymbol{\Pi}^j \boldsymbol{Z}_l^T\right)^{-1}$. $\boldsymbol{\Pi}^j$ is introduced in Sec 4.1. Since we consider the binary sensitive attribute, $j$ is set as 0 without losing generality and is omitted in the notations as $\boldsymbol{\pi}^j(\cdot) := \boldsymbol{\pi}(\cdot)$. After considering all the latent representations, the coherence loss objective is defined as the summation of all coherence over all layers, i.e.,

$$\mathscr{L}_{Coh} = \sum_{l=0}^{L} Coh(\boldsymbol{Z}_l') = \sum_{l=0}^{L} \widehat{Var}\left(\boldsymbol{\pi}(\boldsymbol{Z}_l')\right). \tag{13}$$

**Prediction loss for GNN training.** The GNNs model is trained on distilled graph $\mathcal{G}' = \{\boldsymbol{A}', \boldsymbol{X}', \boldsymbol{Y}'\}$ with prediction loss. We adopt $L$-layer GNNs model, where $\theta$ is the parameter of the GNN. We also adopt cross-entropy loss by default:

$$\mathscr{L}_{CE} = \mathcal{L}\left(\text{GNN}_\theta\left(\boldsymbol{A}', \boldsymbol{X}'\right), \boldsymbol{Y}'\right), \tag{14}$$

### 4.3 Final Objective and Training Algorithm

**Outer loop optimization.** In the outer loop, we optimize the fair graph synthesizer with gradient matching loss and coherence loss:

$$\min_{\boldsymbol{X}', \boldsymbol{A}'} \mathscr{L}_{GM} + \alpha \mathscr{L}_{Coh}, \tag{15}$$

where $\alpha$ is a hyperparameter to regularize the debiasing intensity. The distilled node attribute $\boldsymbol{X}'$ and the distilled node label $\boldsymbol{Y}'$ are initialized with the nodes uniformly sampling from real graph data $\mathcal{G}$.

**Inner loop optimization.** The GNN parameter $\theta$ is optimized in the inner loop:

$$\min_\theta \mathscr{L}_{CE}\left(\text{GNN}_\theta\left(\boldsymbol{A}', \boldsymbol{X}'\right), \boldsymbol{Y}'\right). \tag{16}$$

Instead of using the real graph data $\mathcal{G}$ to calculate the loss, we use the distilled graph $\mathcal{G}'$. It empirically shows good performance and better efficiency. But the adversarial training baseline uses $\mathcal{G}$ as it needs the sensitive attribute for discriminator training.

Table 1: Comparison on utility and bias mitigation between GNNs with real graph data (denoted as Real), the distilled small graph without debiasing (denoted as Vanilla), and debiased distilled graph (denoted as FGD) as input. ↑ denotes the larger, the better; ↓ denotes the opposite. The best ones are in **bold**. The better performers in Vanilla and FGD are underlined.

| | | GCN | | | SGC | | | GraphSAGE | | |
|---|---|---|---|---|---|---|---|---|---|---|
| | | Real | Vanilla | FGD | Real | Vanilla | FGD | Real | Vanilla | FGD |
| Pokec-z | ACC ↑ | **70.96±0.4%** | 66.36±1.0% | 66.58±0.7% | **70.79±0.1%** | 68.31±0.7% | 68.36±0.3% | **70.59±0.3%** | 67.13±0.4% | 67.83±0.6% |
| | AUC ↑ | **78.19±0.2%** | 70.30±0.4% | 70.48±0.3% | **77.16±0.0%** | 73.51±0.6% | 72.92±0.4% | **77.91±0.2%** | 70.47±0.0% | 70.26±0.5% |
| | F1 ↑ | **72.16±0.5%** | 65.66±0.7% | 66.48±0.8% | **71.21±0.0%** | 68.09±0.4% | 67.94±0.6% | **72.07±0.4%** | 66.34±0.8% | 66.62±0.7% |
| | $\Delta_{DP}$ ↓ | 4.13±1.3% | 2.84±1.1% | **1.75±1.1%** | 4.64±0.1% | 7.60±1.7% | 5.77±0.2% | 4.54±1.3% | 3.74±0.8% | **2.17±1.6%** |
| | $\Delta_{EO}$ ↓ | 4.57±1.7% | 2.19±1.3% | **1.19±1.0%** | 5.26±0.1% | 7.88±1.9% | **4.78±0.2%** | 5.25±1.2% | **2.50±1.2%** | 2.56±1.2% |
| Pokec-n | ACC ↑ | **71.97±0.3%** | 50.10±2.7% | 54.80±1.5% | **71.16±0.0%** | 68.06±0.9% | 68.19±0.8% | **71.91±0.3%** | 52.80±2.2% | 58.40±1.6% |
| | AUC ↑ | **78.15±0.2%** | 51.09±2.3% | 63.75±0.5% | **76.34±0.0%** | 69.96±0.3% | 70.18±0.5% | **77.56±0.1%** | 53.95±2.8% | 60.06±2.3% |
| | F1 ↑ | **69.92±0.4%** | 44.21±4.6% | 49.90±5.1% | **67.63±0.0%** | 63.95±0.1% | 64.03±0.1% | **70.01±0.3%** | 58.75±6.1% | 62.36±2.0% |
| | $\Delta_{DP}$ ↓ | **0.59±0.4%** | 4.02±0.6% | 0.66±0.5% | 4.3±0.1% | 5.00±0.5% | 4.7±0.5% | **0.99±0.4%** | 2.38±2.8% | 2.09±1.6% |
| | $\Delta_{EO}$ ↓ | **1.04±0.6%** | 5.20±1.0% | 1.20±1.2% | 2.26±0.1% | 4.6±0.9% | 4.26±1.2% | **1.64±0.6%** | 2.81±3.0% | 2.02±1.2% |
| German | ACC ↑ | 74.37±0.4% | 72.83±0.8% | 70.50±0.1% | 72.62±1.6% | 70.24±0.2% | 70.06±0.1% | 74.24±0.2% | 72.00±0.5% | 71.43±0.9% |
| | AUC ↑ | 74.31±0.2% | 57.75±4.8% | 57.89±5.5% | 74.94±1.2% | 54.82±0.4% | 53.92±1.4% | 71.37±0.4% | 57.32±0.4% | 58.76±5.9% |
| | F1 ↑ | 84.24±0.1% | 83.51±0.4% | 83.09±0.4% | 83.13±0.3% | 82.43±0.0% | 82.36±0.0% | 84.18±0.1% | 83.12±0.4% | 82.93±0.3% |
| | $\Delta_{DP}$ ↓ | 4.8±3.9% | 5.38±3.3% | **1.93±0.1%** | 3.36±2.8% | 2.10±2.8% | **1.3±0.9%** | 3.00±0.8% | 5.33±2.9% | **0.76±0.5%** |
| | $\Delta_{EO}$ ↓ | 2.50±2.7% | 1.04±1.1% | **0.61±0.4%** | 9.38±9.0% | 2.44±0.1% | **0.28±0.2%** | 1.64±0.5% | 2.89±1.0% | **0.52±0.4%** |
| Credit | ACC ↑ | 80.54±0.0% | 76.92±1.9% | 77.91±0.1% | 79.66±0.1% | 77.41±0.0% | 77.36±0.7% | 80.52±0.0% | 77.91±0.5% | 77.86±0.1% |
| | AUC ↑ | 75.89±0.0% | 68.82±2.1% | 68.87±0.2% | 73.39±0.1% | 71.78±0.0% | 72.02±0.1% | 75.89±0.0% | 71.12±0.2% | 71.36±0.1% |
| | F1 ↑ | 88.41±0.0% | 85.47±2.0% | 87.33±0.4% | 88.09±0.0% | 85.46±0.0% | 85.37±0.7% | 88.41±0.0% | 87.00±0.9% | 86.79±0.5% |
| | $\Delta_{DP}$ ↓ | 5.41±0.6% | 12.04±5.4% | **4.94±4.8%** | 2.78±0.3% | 9.58±0.4% | 7.28±2.8% | 6.22±0.6% | 8.77±2.0% | **4.15±0.8%** |
| | $\Delta_{EO}$ ↓ | 3.12±0.6% | 9.58±5.5% | 3.56±3.4% | 1.19±0.3% | 7.14±0.5% | 5.56±0.1% | 3.92±0.5% | 6.72±4.0% | 3.34±0.9% |
| Recidivism | ACC ↑ | 94.45±0.0% | 70.89±1.8% | 70.09±2.6% | 85.10±0.1% | 73.32±0.2% | 73.10±0.5% | 94.48±0.0% | 73.66±1.0% | 70.63±2.0% |
| | AUC ↑ | 97.76±0.0% | 71.95±2.8% | 75.81±4.3% | 92.24±0.1% | 72.23±0.6% | 72.44±0.1% | 97.78±0.0% | 75.84±1.2% | 70.47±5.7% |
| | F1 ↑ | 92.32±0.0% | 55.30±3.9% | 52.97±8.3% | 76.94±0.3% | 60.77±0.8% | 60.45±0.6% | 92.35±0.1% | 60.87±2.4% | 56.39±6.1% |
| | $\Delta_{DP}$ ↓ | 6.52±0.1% | 2.68±1.0% | **0.54±0.3%** | 7.89±0.0% | 2.85±0.9% | **1.08±0.7%** | 6.61±0.1% | 0.97±0.8% | **0.10±0.1%** |
| | $\Delta_{EO}$ ↓ | 3.45±0.4% | 1.41±0.5% | **0.46±0.2%** | 8.55±0.2% | 2.69±0.6% | 1.32±0.8% | 3.56±0.3% | 0.93±0.8% | **0.48±0.4%** |

## 5 Experiments

We design experiments to validate the effectiveness of the proposed framework FGD and answer the following research questions: **RQ.1** How well can FGD mitigate the bias in the distilled graph and alleviate the fairness issue of the GNNs trained on the distilled small graph? **RQ.2** How well can FGD balance the trade-off between accuracy and bias mitigation compared with other debiasing baselines? **RQ.3** Can FGD further improve the utility or bias mitigation as an add-on module to other bias mitigation methods?

### 5.1 Experimental setting

**Datasets.** We use five real-world datasets, including Pokec-z, Pokec-n [Dai and Wang, 2021, Takac and Zabovsky, 2012], German, Credit, and Recidivism [Agarwal et al., 2021]. The detailed setting of the datasets is in Appendix F

**GNN Models.** We adopt three popular GNN variants in our experiments, including GCN [Kipf and Welling, 2016], SGC [Wu et al., 2019], GraphSAGE [Hamilton et al., 2017].

**Baselines.** Given the absence of fair graph distillation work, we compare our approach with four baselines. (1) *Real* uses real graph data to train a GNN model. (2) *Vanilla* applies a vanilla graph distillation algorithm [Jin et al., 2022] for distilled graph data learning and GNN model training. (3) *FairGNN*[Dai and Wang, 2021] trains a fair GNN model adversarially on real graph data, aiming to achieve good prediction while fooling a discriminator. (4) *EDITS* [Dong et al., 2022] is a model-agnostic debiasing method that rewires graph data for fair GNNs.

**Evaluation Metrics.** We evaluate model performance from model utility and bias measurements. Good performance represents low bias and high model utility. For model utility metrics, we adopt accuracy (ACC), the area under the receiver operating characteristic curve (AUC), and F1 score to measure prediction performance. For bias measurement, we adopt two commonly used fairness metrics, i.e., demographic parity ($\Delta_{DP}$) and equal opportunity ($\Delta_{EO}$) [Beutel et al., 2017, Louizos et al., 2015]. Denote the binary label as $y \in \{0, 1\}$, and sensitive attribute as $s \in \{0, 1\}$. $\hat{y} \in \{0, 1\}$ denotes the model prediction. The violation of DP and EO are given by $\Delta_{DP} = |P(\hat{y} = 1 \mid s = 0) - P(\hat{y} = 1 \mid s = 1)|$, and $\Delta_{EO} = |P(\hat{y} = 1 \mid y = 1, s = 0) - P(\hat{y} = 1 \mid y = 1, s = 1)|$.

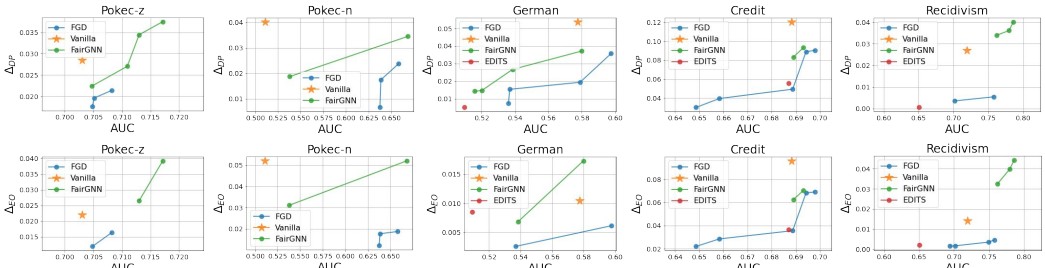

Figure 4: Trade-off comparison between FGDand other baselines for five real-world graph datasets.

## 5.2 Debiasing distilled Graph

In response to **RQ.1**, we assess FGD's bias mitigation and prediction performance across various GNN architectures and datasets, as shown in Table 1. We compare the $\Delta_{DP}$ and $\Delta_{EO}$ values of GNNs trained on real graphs (*Real*), vanilla distilled graphs (*Vanilla*), and debiased distilled graphs via FGD (*FGD*). Key findings include: (1) Models trained on *Real* graphs consistently outperform *Vanilla* and *FGD* in utility, though *FGD*'s utility matches or surpasses *Vanilla*. (2) FGD consistently yields lower bias than *Vanilla*, and outperforms *Real* on 4 out of 5 datasets, excluding Poken-n.

We compare the coherence bias of distilled graphs generated by *Vanilla* and *FGD* methods across five real-world datasets with the GCN architecture (Table 2). Our analysis reveals that FGD reduces unfairness, reflected in lower coherence bias in the distilled graphs. This consistency confirms the effectiveness of coherence bias as a measure of distilled graph bias.

## 5.3 Trade-Off Comparison

In response to **RQ.2**, we compare the trade-off between model utility and bias mitigation against other baselines using the GCN architecture. We utilize the Pareto frontier Ishizaka and Nemery [2013] to evaluate our approach's utility-fairness trade-off, using different hyperparameters. The Pareto frontier graphically represents optimal trade-offs in multi-objective optimization. We use AUC as the utility metric and $\Delta_{DP}$ and $\Delta_{EO}$ as fairness metrics. Higher AUC and lower $\Delta_{DP}/\Delta_{EO}$ are preferred, so models with Pareto frontier curves closer to the bottom right corner (AUC on the horizontal axis and $\Delta_{DP}/\Delta_{EO}$ on the vertical) have better trade-off performance.

Table 2: Coherence bias comparison between vanilla distilled graph (denoted as Vanilla) and fair distilled graph (denoted as FGD). The lower, the better. The best ones are marked in bold. The architecture model is GCN.

|  | Vanilla | FGD |
|---|---|---|
| Pokec-z | 0.009468 | **0.002123**$(-77.57\%)$ |
| Pokec-n | 0.004464 | **0.000432**$(-90.32\%)$ |
| German | 0.012772 | **0.003489**$(-72.68\%)$ |
| Credit | 0.011864 | **0.002866**$(-75.84\%)$ |
| Recidivism | 0.000098 | **0.000038**$(-61.22\%)$ |

Figure 4 shows the results for models trained on the real graph, the distilled graph debiased by baseline methods (vanilla graph distillation, FairGNN, and EDITS[5]) and the distilled graph debiased by FGD. We can observe: (1) From a model utility perspective, FGD performs comparably to other baselines, like vanilla graph distillation, FairGNN, and EDITS [6], suggesting it preserves sufficient information for node classification. (2) In terms of bias mitigation, all baselines show effectiveness, with FGD exhibiting the best results. (3) When considering the utility-fairness trade-off, FGD's Pareto front curve lies at the bottom right corner of all baselines, signifying it offers the best balance. Thus, FGD outperforms other baselines in balancing model utility and bias mitigation.

## 5.4 Add-on Module

---

[5]EDITS publishes fair graph for German, Credit and Recidivism dataset on Github

[6]Out of memory (OOM) issue appears when running EDITS on Pokec-z and Pokec-n datasets.

In addition to its superior trade-off performance, our method, FGD, can enhance other debiasing baselines like FairGNN and EDITS by acting as an add-on debias module. This compatibility is due to the fact that these baselines can replace the cross-entropy loss in the GNN training module. To answer **RQ.3**, we conducted experiments on the Credit dataset comparing FairGNN/EDITS performance with and without FGD. As shown in Figure 5, FairGNN/EDITS coupled with FGD delivers better utility-fairness

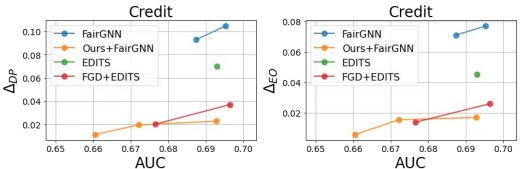

Figure 5: Trade-off comparison between FairGNN, EDITS and FairGNN+FGD, EDITS+FGD on Credit dataset.

trade-off, demonstrating FGD's potential to boost other debias methods.

## 6 Related Work

**Dataset Distillation** & **Knowledge Distillation.** Dataset Distillation (DD) and Knowledge Distillation (KD) are methods to improve the efficiency of training deep neural networks. DD synthesizes a small dataset encapsulating the knowledge of a larger one, achieving comparable model performance [Wang et al., 2018, Kim et al., 2022, Lee et al., 2022, Zhao et al., 2021a, Yang et al., 2022]. It employs a bi-level optimization approach, with dataset condensation (DC) speeding up the process via gradient matching of model parameters. DD also helps with repeated training or privacy applications like continual learning, neural architecture search, and privacy-preserving scenarios. Meanwhile, graph data condensation methods have been developed for node and graph classification tasks [Jin et al., 2022]. KD, on the other hand, enhances computational efficiency through model compression and acceleration. It trains a compact student model using the knowledge from a larger teacher model Gou et al. [2021]To address the scarcity and high complexity of labeled data in GNNs, knowledge distillation (KD) was introduced to enhance existing GNNs Liu et al. [2023a], Wang et al. [2023], also for fairness problem Dong et al. [2023]. While KD focuses on model compression, DD targets data compression, each improving efficiency from model-centric and data-centric perspectives.

**Fair Graph Learning.** Fairness in machine learning has attracted many research efforts[Chuang and Mroueh, 2020, Zhang et al., 2018, Du et al., 2021, Jiang et al., 2022b, Han et al., 2023, Jiang et al., 2023]. Many technologies are introduced in graph neural networks to achieve fair graph learning in node classification tasks, including optimization with regularization [Jiang et al., 2022a], rebalancing [Zeng et al., 2021], adversarial learning [Dai and Wang, 2021, Bose and Hamilton, 2019, Fisher et al., 2020] and graph rewiring [Köse and Shen, 2021, Dong et al., 2022]. For link prediction, dyadic fairness and corresponding graph rewiring solutions are also developed in [Li et al., 2021]. Another line of work focuses on solving the individual fairness problem on the graph data Song et al. [2022], Dong et al. [2021], Kang et al. [2020].

## 7 Conclusion

Despite the ability of graph distillation to condense valuable graph data, this study finds that the vanilla method can worsen fairness issues. Therefore, we introduce a fair graph distillation process to generate fair distilled graph data. As the distilled graph lacks the nodes' sensitive attributes, conventional fair methods cannot be directly applied. However, we identify a consistent geometric phenomenon in graph distillation to estimate these sensitive attributes. We also introduce a new bias metric, coherence, and propose a bi-level optimization framework, FGD, for fair graph distillation. Experimental results validate FGD's effectiveness in mitigating bias while maintaining model utility across various GNN architectures and datasets. Future work will focus on addressing individual fairness issues and non-binary sensitive attribute conditions, among other aspects, as discussed in Appendix H.

## Acknowledgements

We are deeply grateful to the National Science Foundation for their unwavering support. This research was substantially facilitated by the funding from grants IIS-1939716 and IIS-1900990.

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

# A   Training Algorithm

The training algorithm for fair graph distillation is shown in Algorithm 1.

---

**Algorithm 1** Fair Graph Distillation

---

**Input:** Training graph data $\mathcal{G} = \{\boldsymbol{X}, \boldsymbol{A}, \boldsymbol{S}, \boldsymbol{Y}\}$, hyperparameters $\alpha$, temperature $\gamma$, number of alternative optimization step $T_{alt}$, distilled label $\boldsymbol{Y}'$.

Initialize $\boldsymbol{X}'$ based on real attributes, synthesizer model $\phi$, and GNNs model $\theta$.

**for** t = 1 to $T_{alt}$ **do**

1. Train GNNs model using distilled graph $\mathcal{G}'_t$ and Equation (15) to obtain $GNN_{\theta_t}$.

2. Given GNNs model $GNN_{\theta_t}$, calculate the gradient distance $D\big(\nabla_\theta \mathcal{L}(\mathcal{G}), \nabla_\theta \mathcal{L}(\mathcal{G}'_t)\big)$ over the real graph $\mathcal{G}$ and distilled graph $\mathcal{G}'_t$.

3. Calculate coherence loss based on GNNs model $GNN_{\theta_t}$, real graph $\mathcal{G}$ and distilled graph $\mathcal{G}'_t$.

4. Train synthesizer model using prediction loss as Equation (16).

**end for**

**Output:** The fair distilled graph $\mathcal{G}' = \{\boldsymbol{A}', \boldsymbol{X}', \boldsymbol{Y}'\}$.

---

# B   Proof of Theorem 3.4

We consider GNNs model to learn node presentation $\boldsymbol{z}_i$ in the real graph $\mathcal{G}$ and then followed a linear classifier $\boldsymbol{W} = [\boldsymbol{w}_0, \cdots, \boldsymbol{w}_{C-1}]$ and softmax layer, where $\boldsymbol{w}_j$ is the weight vector connected to the $j$-tj output neuron. We first focus on the relation between the latent representation and the gradient of the linear classification layer. It is easy to obtain the cross-entropy loss $J_i$ ($J'_i$) for $i$-th node with label $y_i$ in real graph $\mathcal{G}$ (distilled graph $\mathcal{G}'$) as follows:

$$J_i = -\log \frac{\exp(\boldsymbol{w}_{y_i}^\top \cdot \boldsymbol{z}_i)}{\sum_k \exp(\boldsymbol{w}_k^\top \cdot \boldsymbol{z}_i)}, \tag{17}$$

Then we define gradient over weight vector as $\boldsymbol{g}_{i,j} = \frac{\partial J_i}{\partial \boldsymbol{w}_j}$ and $\boldsymbol{g}'_{i,j} = \frac{\partial J'_i}{\partial \boldsymbol{w}_j}$ in the real and distilled graph. If $j = y_i$, we can obtain

$$
\begin{aligned}
\boldsymbol{g}_{i,y_i} &= -\frac{\sum_k \exp(\boldsymbol{w}_k^\top \cdot \boldsymbol{z}_i)}{\exp(\boldsymbol{w}_{y_i}^\top \cdot \boldsymbol{z}_i)} \\
&\quad \cdot \frac{\exp(\boldsymbol{w}_{y_i}^\top \cdot \boldsymbol{z}_i) \sum_k \exp(\boldsymbol{w}_k^\top \cdot \boldsymbol{z}_i) - \exp^2(\boldsymbol{w}_{y_i}^\top \cdot \boldsymbol{z}_i)}{\big(\sum_k \exp(\boldsymbol{w}_k^\top \cdot \boldsymbol{z}_i)\big)^2} \cdot \boldsymbol{z}_i \\
&= -\boldsymbol{z}_i + \frac{\exp(\boldsymbol{w}_{y_i}^\top \cdot \boldsymbol{z}_i)}{\sum_k \exp(\boldsymbol{w}_k^\top \cdot \boldsymbol{z}_i)} \cdot \boldsymbol{z}_i,
\end{aligned}
\tag{18}
$$

Similarly, for $j \neq y$, we have

$$\boldsymbol{g}_{i,j} = \frac{\exp(\boldsymbol{w}_{y_i}^\top \cdot \boldsymbol{z}_i)}{\sum_k \exp(\boldsymbol{w}_k^\top \cdot \boldsymbol{z}_i)} \cdot \boldsymbol{z}_i, \tag{19}$$

In other words, the gradient of the loss for $i$-th node with label $y_i$ with respect to the weight vector connected to the $j$-th output neuron is given by

$$\boldsymbol{g}_{i,j} = \frac{\exp(\boldsymbol{w}_{y_i}^\top \cdot \boldsymbol{z}_i)}{\sum_k \exp(\boldsymbol{w}_k^\top \cdot \boldsymbol{z}_i)} \cdot \boldsymbol{z}_i - \mathbb{1}_{j=y_i} \boldsymbol{z}_i. \tag{20}$$

Based on Assumption 3.1, each model parameter in the last softmax layer satisfies the same distribution. In other words, the expectation of all predictions are the same, i.e.,

$$\mathbb{E}_{\mathcal{P}_\theta}\Big[\frac{\exp(\boldsymbol{w}_0^\top \cdot \boldsymbol{z}_i)}{\sum_k \exp(\boldsymbol{w}_k^\top \cdot \boldsymbol{z}_i)}\Big] = \cdots = \mathbb{E}_{\mathcal{P}_\theta}\Big[\frac{\exp(\boldsymbol{w}_{C-1}^\top \cdot \boldsymbol{z}_i)}{\sum_k \exp(\boldsymbol{w}_k^\top \cdot \boldsymbol{z}_i)}\Big]. \tag{21}$$

Note that the gradient calculation is based on backpropagation, the gradient for the last linear classification layer is quite critical for the gradient of other layers. Hence we consider the gradient of the last linear classification layer in the real graph, shown by

$$
\begin{aligned}
\mathbb{E}_{\theta \sim \mathcal{P}_\theta}\big[\nabla_{\boldsymbol{w}_j}\mathcal{L}(\mathcal{G})\big] &= \mathbb{E}_{\theta \sim \mathcal{P}_\theta}\big[\frac{1}{N}\sum_{i=1}^{N}\boldsymbol{g}_{i,j}\big] \\
&= \frac{1}{NC}\sum_{i=1}^{N}\boldsymbol{z}_i - \frac{1}{N}\sum_{\{i:y_i=j\}}\boldsymbol{z}_i,
\end{aligned}
\tag{22}
$$

Similarly, we have the gradient of the last linear classification layer in the distilled graph as follows:

$$
\begin{aligned}
\mathbb{E}_{\theta \sim \mathcal{P}_\theta}\big[\nabla_{\boldsymbol{w}_j}\mathcal{L}(\mathcal{G}')\big] &= \mathbb{E}_{\theta \sim \mathcal{P}_\theta}\big[\frac{1}{N'}\sum_{i=1}^{N}\boldsymbol{g}'_{i,j}\big] \\
&= \frac{1}{N'C}\sum_{i=1}^{N'}\boldsymbol{z}'_i - \frac{1}{N'}\sum_{\{i:y'_i=j\}}\boldsymbol{z}'_i,
\end{aligned}
\tag{23}
$$

Under assumption 3.2, it is easy to know that the optimal solution to minimizing the objective $\min_{\mathcal{G}'}\mathbb{E}_{\theta \sim \mathcal{P}_{\boldsymbol{W}}}\big[||\nabla_{\boldsymbol{W}}\mathcal{L}(\mathcal{G})-\nabla_{\boldsymbol{W}}\mathcal{L}(\mathcal{G}')||^2\big]$ satisfy $\nabla_{\boldsymbol{W}}\mathcal{L}(\mathcal{G}) = \nabla_{\boldsymbol{W}}\mathcal{L}(\mathcal{G}')$. Since the distilled label is sampling to keep class label probability, we have $\frac{|\{i:y'_i=j\}|}{N'} = \frac{|\{i:y_i=j\}|}{N}$ for any class index $i$. Therefore, based on Equations (22) and (23), we have the optimal distilled graph satisfy

$$
\frac{1}{N}\sum_{i=1}^{N}\boldsymbol{z}_i = \frac{1}{N'}\sum_{i=1}^{N'}\boldsymbol{z}'_i.
\tag{24}
$$

## C   Proof of Ridge Regression

Define objective function $J = \gamma\|\boldsymbol{z}' - \boldsymbol{Z}_s^\top\boldsymbol{q}\|_2^2 + \|\boldsymbol{q}\|_2^2$, it is easy to obtain

$$
\frac{\partial J}{\partial \boldsymbol{q}} = -2\gamma\boldsymbol{Z}_s\big(\boldsymbol{z}' - \boldsymbol{Z}_s^\top\boldsymbol{q}\big) + 2\boldsymbol{q} = 0
\tag{25}
$$

Therefore, the optimal $\boldsymbol{q}^* = \gamma(\boldsymbol{I} + \gamma\boldsymbol{Z}_s\boldsymbol{Z}_s^\top)^{-1}\boldsymbol{Z}_s\boldsymbol{z}'$. Therefore, the projection of representation $\boldsymbol{z}'$ in the complement space of sensitive group $\boldsymbol{Z}_s$ is given by

$$
\boldsymbol{z}' - \boldsymbol{Z}_s^\top\boldsymbol{q}^* = \boldsymbol{z}' - \gamma\boldsymbol{Z}_s^\top(\boldsymbol{I} + \gamma\boldsymbol{Z}_s\boldsymbol{Z}_s^\top)^{-1}\boldsymbol{Z}_s\boldsymbol{z}'
\tag{26}
$$

## D   More Results on Consistent Span Space

We conduct experiments to measure the distance between $span(Z)$ and $span(Z')$ using principle angles between subspaces and emperically shows that $span(Z) \approx span(Z')$ in the real dataset.

The concept of principal angle is used in linear algebra to measure the similarity between two subspaces of a vector space. It helps quantify how close or far apart these subspaces are. Given subspace, $\mathbf{L}, \mathbf{M} \subseteq \mathbb{R}^n$, with $\dim \mathbf{L} = l \geq \dim \mathbf{M} = m$, there are m principal angles between L and M denoted as $0 \leq \theta_1 \leq \theta_2 \leq \cdots \leq \theta_m \leq \frac{\pi}{2}$ between L and M are recursively defined, where

$$
\cos(\theta_i) := \min\left\{\frac{\langle\mathbf{x},\mathbf{y}\rangle}{\|\mathbf{x}\|\|\mathbf{y}\|} \mid \mathbf{x} \in \mathbf{L}, \mathbf{y} \in \mathbf{M}, \mathbf{x} \perp \mathbf{x}_k, \mathbf{y} \perp \mathbf{y}_k, k = 1, \cdots, i-1\right\}.
\tag{27}
$$

Notably, when the two subspaces are aligned, the principal angels are close to 0. We report the average principal angles of $span(Z)$ and $span(Z')$ on all datasets as following:

- Pokec-z: $1.08 \times 10^{-6}$
- Pokec-n: $1.03 \times 10^{-6}$
- German: $4.84 \times 10^{-7}$

- Credit: $2.57 \times 10^{-7}$
- Recidivism: $3.87 \times 10^{-7}$

In the experiments, the principal angles of $span(Z)$ and $span(Z')$ on all dataset are nearly 0. This indicates that the distance between space $span(Z')$ and space $span(Z)$ are quite small in practice.

Additionally, we would like to mention that [1] provides the rigorous proof of $z' \in span(Z)$ for distribution matching under several assumptions (although we can not prove it under gradient matching setting). According to formulas 21 from [1], it is assumed that (1) the **linear extractor** $\psi_{\boldsymbol{\theta}} : \mathbb{R}^d \to \mathbb{R}^k$ such that $k < d$, $\boldsymbol{\theta} = [\theta_{i,j}] \in \mathbb{R}^{k \times d}$, $\theta_{i,j} \overset{iid}{\sim} \mathcal{N}(0, 1)$ and for an input $\mathbf{z}$, $\psi_{\boldsymbol{\theta}}(\mathbf{x}) = \boldsymbol{\theta}\mathbf{z}$. When using distribution match method for data condensation, we have:

$$\frac{\partial L}{\partial \mathbf{z}'_i} = \frac{\partial E_\theta ||d||^2}{\partial \mathbf{z}'_i} = -\frac{2}{|N'|} \left( \frac{1}{N} \sum_{j=1}^{N} \mathbf{z}_j - \frac{1}{N'} \sum_{j=1}^{N'} \mathbf{z}'_j \right)^T \cdot E[\boldsymbol{\theta}^t \boldsymbol{\theta}]$$

where $d := \theta \left( \frac{1}{N} \sum_{j=1}^{N} \mathbf{z}_j - \frac{1}{N'} \sum_{j=1}^{N'} \mathbf{z}'_j \right)$, $\mathbb{E}\left[ \boldsymbol{\theta}^\top \boldsymbol{\theta} \right] = k\mathbf{I}_d$ by definition of $\boldsymbol{\theta}$, and $\mathbf{I}_d$ is the identity matrix of $\mathbb{R}^d$. the projection components of $span(Z)^\perp$ remain zero throughout the optimization process of DM. And we use $\mathbf{z}_i$ to initialize $\mathbf{z}'_i$, thus $\mathbf{z}' \in span(Z)$. However, in the implementation we use gradient matching instead of distribution matching.

Table 3: Statistical Information on Datasets

| Dataset | # Nodes | # Attributes | # Edges | Avg. degree | Sens | Label |
|---|---|---|---|---|---|---|
| Pokec-n | 6,185 | 59 | 21,844 | 7.06 | Region | Working field |
| Pokec-z | 7,659 | 59 | 29,476 | 7.70 | Region | Working field |
| German | 1,000 | 27 | 21,242 | 44.50 | Gender | Credit status |
| Credit | 30,000 | 13 | 1,436,858 | 95.80 | Age | Future default |
| Recidivism | 18,876 | 18 | 321,308 | 34.00 | Race | Bail decision |

# E  Preliminary Motivation

We have added experiments comparing the fairness performance of various fair GNNs trained on synthetic and real graph data. Specifically, we report the results (using demographic parity (DP), equal opportunity (EO), and individual unfairness (IND) Song et al. [2022] as metrics) with EDITS Dong et al. [2022], FairGNN Dai and Wang [2021], InFoRM Kang et al. [2020], and REDRESS Dong et al. [2021] on five datasets in our paper. EDITS is a pre-processing debiasing method, FairGNN is an in-processing debiasing method, and InFoRM and REDRESS focus on individual fairness. We encountered out-of-memory (OOM) issues when implementing GUIDE and REDRESS on an NVIDIA GeForce RTX A5000 (24GB GPU memory), so we used InFoRM as the baseline. Due to the extensive training time required for REDRESS, we only report results on the German dataset for REDRESS. We use demographic parity (DP), equal opportunity (EO), and individual unfairness (IND) as metrics.Table 4 demonstrates the result. From Table 4, we can see that in terms of the group fairness metrics (DP, EO), the fairness problem becomes uniformly worse on the Credit, German, and Pokecn datasets for all debiasing methods. For the Recidivism dataset, the distilled graph shows fewer fairness issues (lower DP or EO), especially for the EDITS method. This may result from the drop in utility of the model trained on the distilled graph (AUC is too low). As shown in Figure 4 of our paper, FGD can achieve a better performance-fairness trade-off compared to the baselines.

# F  Dataset Statistics

**Pokec.**  The Pokec dataset consists of millions of anonymized user profiles from Slovakia's most popular social network in 2012, with information such as gender, age, hobbies, interests, education, and working field. The dataset was sampled into Pokec-z and Pokec-n based on user province, with region as the sensitive attribute. The task is to predict user working field.

Table 4: Utility and group fairness comparison between real graph and distilled graph with various debias method. **Bold** value indicates worse fairness performance.

| | | Recidivism | | Credit | | German | | Pokecn | | Pokecz | |
|---|---|---|---|---|---|---|---|---|---|---|---|
| | | Real | Distillated | Real | Distillated | Real | Distillated | Real | Distillated | Real | Distillated |
| EDITS | AUC↑ | 0.971 | 0.658 | 0.740 | 0.704 | 0.668 | 0.506 | OOM | OOM | OOM | OOM |
| | DP↓ | 0.067 | 0.005 | 0.027 | **0.063** | 0.009 | **0.024** | OOM | OOM | OOM | OOM |
| | EO↓ | 0.038 | 0.011 | 0.018 | **0.028** | 0.008 | **0.030** | OOM | OOM | OOM | OOM |
| FairGNN | AUC↑ | 0.977 | 0.788 | 0.759 | 0.720 | 0.742 | 0.645 | 0.782 | 0.676 | 0.784 | 0.723 |
| | DP↓ | 0.065 | 0.046 | 0.062 | **0.123** | 0.010 | **0.013** | 0.005 | **0.044** | 0.042 | **0.037** |
| | EO↓ | 0.037 | 0.046 | 0.037 | **0.091** | 0.001 | **0.011** | 0.006 | **0.062** | 0.051 | **0.038** |
| InFoRM | AUC↑ | 0.906 | 0.708 | 0.741 | 0.717 | 0.642 | 0.538 | 0.743 | 0.644 | 0.751 | 0.708 |
| | DP↓ | 0.011 | 0.118 | 0.004 | **0.174** | 0.085 | 0.018 | 0.009 | 0.009 | 0.020 | **0.048** |
| | EO↓ | 0.024 | 0.092 | 0.001 | **0.135** | 0.153 | 0.017 | 0.013 | 0.015 | 0.018 | **0.038** |
| | IND↓ | 8098 | **3596022** | 2699 | **338149** | 4360 | **24888** | 6466 | **272013** | 6828 | **199853** |
| REDRESS | AUC↑ | OOM | OOM | OOM | OOM | 0.719 | 0.483 | OOM | OOM | OOM | OOM |
| | DP↓ | OOM | OOM | OOM | OOM | 0.005 | **0.043** | OOM | OOM | OOM | OOM |
| | EO↓ | OOM | OOM | OOM | OOM | 0.010 | **0.073** | OOM | OOM | OOM | OOM |
| | IND↓ | OOM | OOM | OOM | OOM | 9728 | **186366** | OOM | OOM | OOM | OOM |

Table 5: Parameter study of $\alpha$. All the value is in scale of $\times 10^2$.

| $\alpha$ | AUC | $\Delta_{DP}$ | $\Delta_{EO}$ | Bias |
|---|---|---|---|---|
| 0.04 | 74.75 | 0.84 | 0.88 | 0.19 |
| 0.5 | 69.42 | 0.66 | 0.43 | 0.15 |
| 0.6 | 69.37 | 0.58 | 0.16 | 0.14 |
| 1.0 | 65.35 | 0.00 | 0.00 | 0.11 |

**German.** The German Graph credit dataset has 1,000 client records with attributes like Gender and LoanAmount, used to classify individuals as good or bad credit risks. The similarity between node attributes is calculated using Minkowski distance and nodes are connected if the similarity is 80% of the maximum similarity.

**Credit.** Credit dataset, consisting of 30,000 individuals with features such as education, credit history, age, and derived spending and payment patterns. The similarity between two node attributes is calculated using Minkowski distance as the similarity measure and the credit defaulter graph network is constructed by connecting nodes with a similarity of 70% of the maximum similarity between all nodes.

**Recidivism.** The US state court bail outcome dataset (1990-2009) contains 18,876 defendant records with past criminal records, demographic attributes, etc. The similarity between node attributes is calculated using Minkowski distance and nodes are connected if the similarity is 60% of the maximum similarity.

# G   More Experimental Details

## G.1   Parameter Study

Here we aim to study the sensitivity of FGD w.r.t. hyper-parameters. Specifically, we show the parameter study of $\alpha$ on Recidivism dataset. Here $\alpha$ controls the intensity to regularize the coherence bias of the distilled small graph. The results in Table 5 indicate that $\alpha$ can control the debiasing and utility performance of the distilled small graph.

## G.2   Implementation Details

**Synthesizer training.** We adopt Adam optimizer for synthesizer training with 0.0002 learning rate. MLP$_\phi$ consists of 3 linear layer with 128 hidden dimension. The outer loop number is 16 while the inner loop is 4 for each epoch. For each experiment, we train with a maximum of 1200 epochs and 3 independent runs. The temperature parameter $\gamma$ is set to 10. $\boldsymbol{X}'$ and $\phi$ are optimized alternatively.

**GNN training.** We adopt Adam optimizer for GNN training with 0.005 learning rate. All GNN models are 2 layers with 256 hidden dimensions. For Pokec-z, Pokec-n, German, Credit, and Recidivism the training epochs are 1500, 1500, 4000, 1000, and 1000 respectively.

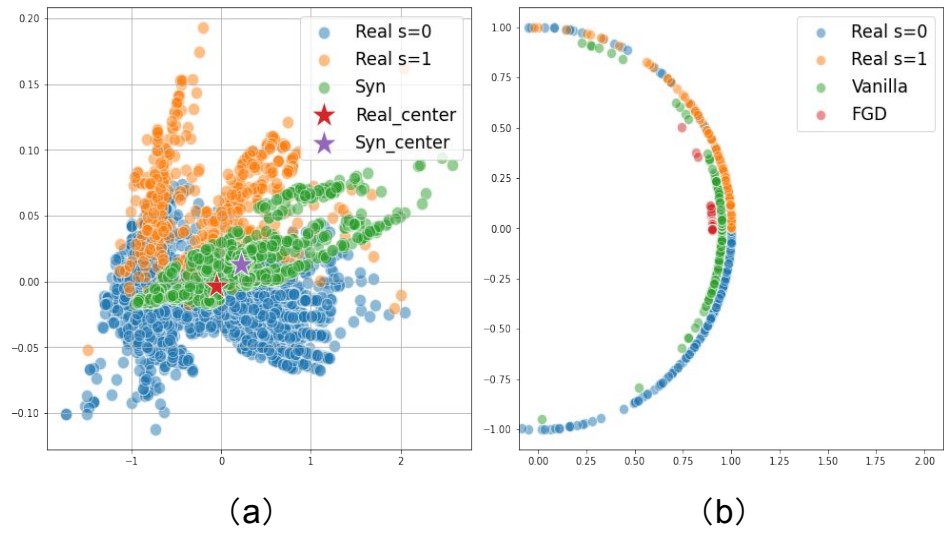

Figure 6: (a) shows the visualization of node representations from real graph and distilled graph, as well as their barycenter, on Credit dataset, after PCA. (b) shows the visualization of geometric intuition of node from real graph and distilled graph on Credit dataset.

### G.3 More Visualization

We also visualize the node representation using PCA. We could observe that the barycenter of node from real graph and distilled graph is very close. And The distribution of node representation after being normalized to the circumference is consistent with the geometric intuition shown in Figure 6.

## H Limitations and Future Work

### H.1 Non-binary Sensitive Attribute

For categorical sensitive attributes, if only one sensitive membership group's embeddings are far away from others, then the mean embeddings will still be close to the majority embeddings, especially for many categories, resulting in low variance (coherence). We argue that only this group with distant embedding (a small portion of samples) can have their sensitive attribute detected using embedding distributions. From a metric perspective, if we adopt the maximized $\Delta_{DP}$ over any sensitive attribute group pair, the bias should be large due to considering the worst case. The proposed coherence may not work well in this scenario, and an advanced coherence can be developed for this case, e.g., the maximized variance over any sensitive group pair. We leave the advanced coherence development for categorical, multiple, or even continuous sensitive attributes in future work.

### H.2 Individual Fairness

From Table 4, we find that all datasets suffer from a surprisingly more severe individual fairness problem (much higher IND score) when the model is trained on the distilled graph, even if we use InFoRM or REDRESS. This could be an interesting direction for future work, and we will add discussion with references in the related work section.

### H.3 Other Tasks

Our paper mainly focuses node classification tasks and it is possible to extend our method to other tasks or other group fairness problems. For instance, FGD may alleviate group fairness issues in link prediction tasks by reducing the coherence bias among different link groups. Exploring other tasks (e.g., recommendation, graph classification) or other fairness metrics (e.g., individual fairness, rank fairness) could be interesting for future work.

