# OpenReview forum: "Fair Graph Distillation"
_NeurIPS.cc/2023/Conference — NeurIPS 2023 poster_

### Official Review · Reviewer_jsrW · 2023-06-14

**Soundness:** 3 good
**Presentation:** 4 excellent
**Contribution:** 3 good
**Rating:** 6
**Confidence:** 3

**Summary:**

This paper proposes fair graph distillation (FGD), as an advanced graph distillation approach to generate fair distilled graphs. FGD focuses on the group fairness issue in graph distillation methods and aims to generate fair distilled graphs with respect to sensitive attributes for nodes. This paper proposes a simple yet effective metric for measuring the bias in representation space, namely coherence, for distilled graphs, and a bi-level optimization framework to generate fair graph distillations iteratively. Experimental results illustrate that the proposed methods can achieve  performance-fairness trade-offs across various datasets and architectures.

**Strengths:**

- This paper considers group fairness issue, which is novel in graph distillation.
- The proposed metric for bias measurement is simple yet convincing.
- The theorems and algorithms introduced in paper is well presented.
- Experimental results show the effectiveness of FGD in debiasing distillation.

**Weaknesses:**

- More baselines in graph distillation are needed. For example, *FairGNN* and *EDITS* are introduced as baselines in debiasing, while no graph distillation methods (such as [1]) are compared.
- Results in Table 1 shows that distilled graphs could achieve some improvements in debiasing, while the AUC performance drops significantly (e.g. from 94% to 70%), which is unacceptable for node classification.


[1] Bo Zhao, Konda Reddy Mopuri, and Hakan Bilen. Dataset Condensation with Gradient Matching. In ICLR 2021

**Questions:**

- Is FGD capable of handling muti-class cases? In the paper authors mentioned to use average coherence to optimize the distilled graph, yet there are no experiments on this.
- What is the computational complexity of the overall algorithm?

**Limitations:**

- The distilled graphs can't achieve comparable performance compared with original graphs.
- The presented method is only tested in binary classification scenario.

---

> ### Author Rebuttal · Authors · 2023-08-08
>
> # 1. More baselines in graph distillation are needed.
> The vanilla baseline we utilized in our work is derived from the graph data condensation method introduced in [2] tailored for graph data using gradient matching, which is the same as [1]. Thus, [1] can not be directly adopted in graph data due to the lack of distilled adjacency matrix. We only choose [2] as our only vanilla graph data distillation baseline.
>
> [1] Bo Zhao, Konda Reddy Mopuri, and Hakan Bilen. Dataset Condensation with Gradient Matching. In ICLR 2021
>
> [2] Jin, Wei, et al. "Graph condensation for graph neural networks." ICLR 2022.
>
> # 2. Utility performance drop.
> Thanks for this great point. We have meticulously tuned the vanilla distillation method for these datasets, taking into consideration the reduction rate. In the original experiments, we used a reduction rate of 0.08 (meaning that the number of nodes in the distilled graph is 8% of the original graph) for all datasets. The small distilled graph leads to significantly worse utilities. Therefore, in the new experiments, we increase the reduction rate to 0.32 and 0.16 to increase the accuracy performance for the Pokec-n and Recidivism datasets, respectively. We report the **updated results on Pokec-n and Recidivism datasets** as follows.
> |            |      |   GCN   |      |
> |:----------:|:----:|:-------:|:----:|
> |            |      | Vanilla |  FGD |
> |            | ACC↑ |   62.16  | 61.84 |
> |            | AUC↑ |   65.82  | 64.59 |
> |   Pokec-n  |  F1↑ |   57.38  | 58.62 |
> |            |  DP↓ |   5.83  | **0.84** |
> |            |  EO↓ |   6.34  | **1.94** |
> |      -      |     -     |        -   |     -    |
> |            | ACC↑ |   78.18  | 77.51 |
> |            | AUC↑ |   75.83  | 75.84 |
> | Recidivism |  F1↑ |   63.84  | 61.98 |
> |            |  DP↓ |   5.25  | **1.34** |
> |            |  EO↓ |   3.49  | **1.98** |
> 1. **Utility Improvement**: The utility performance of the model improves when trained on a larger distilled graph but exhibits worse fairness performance.
> 2. **Utility Gap Attribution**: The utility gap observed is attributable to the distillation method itself.
> 3. **Fairness Issue Persistence**: Even when the utility performance is subpar, the fairness problem still exists and is a concern.
> 4. **Our Method's Advantage**: Compared to the vanilla distillation method, our Fair Graph Distillation (FGD) enhances fairness performance while maintaining or even improving utility performance.
>
> # 3. What is the computational complexity of the overall algorithm?
> We provide the time and space complexity analysis as follows:
>
> 1. **Forward Process of GCN**:
>     - On the original graph time and space complexity: $O\left(r^L N d^2\right)$ and $O\left(r^L N d + N^2\right)$, where $r$ is the number of sampled neighbors per node, $L$ is the number of layers, and $N$ and $d$ are the number of nodes in real data and hidden units, respectively.
>     - On the condensed graph time and space complexity: $O\left(L N^{\prime 2} d + L N' d\right)$ and $O\left(L N^{\prime 2} + L N' d\right)$, where $N'$ the number of nodes in distilled data .
>
> 2. **Backward Propagation**:
>     - The time complexity of calculating the second-order derivatives is an additional $O\big(\left|\boldsymbol{\theta}\right| N'(N'+d)\big)$, where $\left|\boldsymbol{\theta}\right|$ is the number of parameters in $\theta$. The space complexity is $O( |\boldsymbol{\theta}|^2 + N'(N'+d))$.
>
> 3. **Coherence Loss**:
>     - The time and space complexity are $O\left(L N^{\prime 2} d \right)$ and $O\left(L N^{\prime 3} d \right)$, respectively.
>
> We will add this complexity analysis in the revised manuscript.
>
> # 4. The distilled graphs can't achieve comparable performance compared with original graphs.
> We have achieved comparable performance compared with the original graphs via tuning the reduction rate. Please see #2 for detailed response.
>
> # 5. The presented method is only tested in binary classification scenario?
> Thank you for noting this limitation. Extending our approach to categorical, multiple, or continuous sensitive attributes indeed presents new challenges. We have discussed such an extension in **Appendix H.1** and leave it for future work.

---

> > ### Comment · Reviewer_jsrW · 2023-08-16
> >
> > Thank you for the clear explanation and additional experimental results. This is an insightful work and I will change my score.

---

> > > ### Author Response · Authors · 2023-08-18
> > >
> > > Dear Reviewer jsrW,
> > >
> > > I want to extend our heartfelt thanks for recognizing what we've been working on. Your thoughtful advice and encouraging feedback have truly helped us take our work to the next level.
> > >
> > > Best regards,
> > >
> > > Authors

---

### Official Review · Reviewer_5Zjc · 2023-07-03

**Soundness:** 3 good
**Presentation:** 2 fair
**Contribution:** 2 fair
**Rating:** 5
**Confidence:** 3

**Summary:**

This paper aim to address the issue of fairness in graph data distillation, a process that condenses large real graphs into smaller distilled versions for more manageable computation with GNNs. They proposed FGD by introducing a new bias metric called coherence and using a bi-level optimization algorithm, which has shown to provide improved performance-fairness trade-offs in numerous experiments.

**Strengths:**

1.	This paper studies an interesting issue of fairness within distilled graphs, which arises due to the absence of sensitive features.
2.	This paper devised a bias measurement named coherence specifically for distilled graphs, and suggests a framework utilizing this metric to facilitate the realization of fair graph distillation. The theoretical analysis in this paper is well founded.
3.	The authors have conducted a thorough experimental analysis, and the presented results indicate that the proposed framework is adaptable to numerous renowned GNNs. This framework improves the trade-off between prediction performance and fairness across a range of datasets, signifying the framework's effectiveness and wide-ranging applicability.


**Weaknesses:**

1.	Are there any fairness studies in dataset distillation in other fields such as computer vision and natural language processing? If so, there is a lack of discussion comparing the proposed framework with other similar works.
2.	The time and space complexity are not mentioned.
3.	In terms of methodology, this paper primarily uses strategies that were already developed before, and the unique contribution is the incorporation of a new loss term, Overall, the technical contribution of this work seems incremental.
4.	The experimental part is inadequate, lacking integration of the proposed framework with large-scale and advanced GNNs, experiments in large datasets and the extra overhead of proposed framework.



**Questions:**

See the weaknesses.

**Limitations:**

Whether the proposed framework can be applied to graph-level tasks, not just node-level ones.

---

> ### Author Rebuttal · Authors · 2023-08-08
>
> # 1. Are there any fairness studies in dataset distillation in other fields?
> In the existing literature, fairness studies related to dataset distillation in the fields of CV or NLP are not commonly found. The only fair distillation work we can found is [1]n which studies the fairness problem on the data distillation for text classification tasks based on adversarial method.
> To our knowledge, our work is the first paper addressing group fairness issues on **graph distillation**, and we focus on graph data due to the unique challenge as graph is non-euclidian data.
> We will include a section in the revised manuscript to discuss the novelty of our work considerations to other fields.
>
> [1] Han, Xudong, et al. "Towards Fair Dataset Distillation for Text Classification." (SustaiNLP). 2022.
>
> # 2. The time and space complexity are not mentioned.
> We provide the time and space complexity analysis as follows:
>
> 1. **Forward Process of GCN**:
>     - On the original graph time and space complexity: $O\left(r^L N d^2\right)$ and $O\left(r^L N d + N^2\right)$, where $r$ is the number of sampled neighbors per node, $L$ is the number of layers, and $N$ and $d$ are the number of nodes in real data and hidden units, respectively.
>     - On the condensed graph time and space complexity: $O\left(L N^{\prime 2} d + L N' d\right)$ and $O\left(L N^{\prime 2} + L N' d\right)$, where $N'$ the number of nodes in distilled data .
>
> 2. **Backward Propagation**:
>     - The time complexity of calculating the second-order derivatives is an additional $O\big(\left|\boldsymbol{\theta}\right| N'(N'+d)\big)$, where $\left|\boldsymbol{\theta}\right|$ is the number of parameters in $\theta$. The space complexity is $O( |\boldsymbol{\theta}|^2 + N'(N'+d))$.
>
> 3. **Coherence Loss**:
>     - The time and space complexity are $O\left(L N^{\prime 2} d \right)$ and $O\left(L N^{\prime 3} d \right)$, respectively.
>
> We will add this complexity analysis in the revised manuscript.
>
> # 3. The technical contribution of this work seems incremental.
> We respectively disagree with this comment. We clarify that one of the contributions is to find the fairness issue in graph distillation problem, which is interesting and novel. As for our technical contribution (i.e., coherence loss), it is significant and non-trivial. For distilled graph, the sensitive attribute is missing and thus it is challenging to identify the bias for node representation in distilled graph data. To tackle this challenge, we find the geometric connections in data distillation and provide the rationale for sensitive attribute estimation. In this way, the estimated sensitive attribute information can be used to measure the bias for the distilled graph. We believe that the proposed coherence loss is significant and non-trivial.
> # 4. Experiments on large-scale and advanced GNNs. The extra overhead of proposed framework.
> It is infeasible to conduct the experiment for large-scale graph datasets in a fairness community since there is no public larger graph dataset.  The datasets we use in the paper are the most common in the graph fairness field. For large-scale GNN structure, we have conducted additional experiments using **GraphSAINT**, a well-known large-scale GNN method. We also record the 100 epoch training time for the vanilla distillation and FGD:
>
> |            |      |      |   GraphSAINT   |      |
> |:----------:|:----:|------|:-------:|:----:|
> |            |           | Real | Vanilla |  FGD |
> |            | ACC↑ |    70.84  |   64.56  | 63.73 |
> |            | AUC↑ |   78.19   |   70.43  | 71.94 |
> | Pokec-z|  F1↑ |   70.94    |   67.84  | 66.74 |
> |            |  DP↓ |   9.18   |   7.33  | 2.70 |
> |            |  EO↓ |   10.82   |   6.58  | 2.06 |
> |            |  100 epoch time |       |   59.8s  | 143.2s |
> |            |   |      |      |   |
> |            | ACC↑ |   69.74   |   61.45  | 58.34 |
> |            | AUC↑ |   75.19   |   62.48  | 60.84 |
> | Pokec-n|  F1↑ |   69.71   |   53.51  | 50.39 |
> |            |  DP↓ |    3.64  |   5.65  | 3.98 |
> |            |  EO↓ |    1.42  |   2.98  | 2.15 |
> |            |  100 epoch time |       |   55.9s  | 125.0s |
> |            |   |      |      |   |
> |            | ACC↑ |   71.17   |   70.63  | 70.06 |
> |            | AUC↑ |   68.83   |   56.68  | 55.30 |
> | German |  F1↑ |   82.34   |   82.11  | 79.83 |
> |            |  DP↓ |   4.74   |   5.25  | 3.53 |
> |            |  EO↓ |   3.57   |   3.49  | 2.62 |
> |            |  100 epoch time |       |   10.8s  | 23.6s |
> |            |   |      |      |   |
> |            | ACC↑ |   80.73   |   77.82  | 78.83 |
> |            | AUC↑ |   75.18   |   71.59  | 71.82 |
> | Credit |  F1↑ |    88.38  |   85.15  | 86.04 |
> |            |  DP↓ |   8.86   |   12.25  | 6.09 |
> |            |  EO↓ |   5.25   |   11.49  | 2.36 |
> |            |  100 epoch time |       |   67.8s  | 179.8s |
> |            |   |      |      |   |
> |            | ACC↑ |   93.37   |   71.38  | 70.72 |
> |            | AUC↑ |   96.18   |   70.28  | 69.71 |
> | Recidivism |  F1↑ |   90.63   |   58.94  | 61.63 |
> |            |  DP↓ |   6.15    |   3.49  | 2.93 |
> |            |  EO↓ |    4.53  |   2.48  | 1.94 |
> |            |  100 epoch time |       |   48.4s  | 64.9s |
>
> From the results, it is seen that our proposed FGD method maintains competitive utility performance while improving fairness metrics. While there is an increase in training time for FGD compared to the vanilla distillation method, the improvement in fairness metrics demonstrates the effectiveness of our approach in addressing fairness without compromising utility.
>
> # 5. Whether the proposed framework can be applied to graph-level tasks, not just node-level ones.
>
> For graph classification task, to the best of our knowledge, there is not any fairness literature on this task. The reason is that the dataset for graph classification is molecular, and thus there is no sensitive attribute. We have added the discussion on extending our method in link prediction in **Appendix H.3**.

---

> > ### Comment · Reviewer_5Zjc · 2023-08-18
> >
> > Thanks for the clarification. I have raised my score to 5.

---

> > > ### Author Response · Authors · 2023-08-18
> > >
> > > Dear Reviewer 5Zjc,
> > >
> > > Allow me to convey our profound thanks for acknowledging our efforts. Your wise counsel and positive recommendations have unquestionably contributed significantly to enhancing our work's quality.
> > >
> > > With kind regards,
> > >
> > > Authors

---

### Official Review · Reviewer_LkqE · 2023-07-04

**Soundness:** 3 good
**Presentation:** 3 good
**Contribution:** 3 good
**Rating:** 7
**Confidence:** 3

**Summary:**

This paper discovered the fairness problem in the distilled GNN methods and then introduce a fair graph distillation process to generate fair distilled graph data. To propose the algorithm, they also introduce a new bias metric, coherence, and propose a bi-level optimization framework, FGD, for fair graph distillation. Theoretical analysis are provided. Experimental results validate FGD’s effectiveness in mitigating bias while maintaining model utility across various GNN architectures and datasets.

**Strengths:**

1.	Good presentation. This paper is easy and comfortable to read and follow.
2.	Solid theoretical analysis and experimental validation.
3.	This paper is the first paper to solve the fairness problem in distilled GNN lines, which have good novelty and contribution.


**Weaknesses:**

1. Some terminology needs simple descirption. It is better to include some preliminary knowledge or terminology description to make paper more self-included.

**Questions:**

1. What is the full name of FGD？ The full needs to be provided when this word first appears.
2. Why the sensitive attribute S is a diagonal matrix, not a vector with the dimension of number of node? Is it for computation convenience?
3. Some terminology needs simple description, such as span space and barycenter.
4. Is this methods only applicable to node classification task? Can it be extended to link prediction or graph classification?

---

> ### Author Rebuttal · Authors · 2023-08-07
>
> # 1. What is the full name of FGD？ The full needs to be provided when this word first appears.
> The acronym FGD stands for Fair Graph Distillation. We will include the full name in Line 54.
>
> # 2. Why the sensitive attribute S is a diagonal matrix, not a vector with the dimension of number of node? Is it for computation convenience?
> We will revise the sensitive attribute $\mathbf{S}$ as  $\mathbf{s}$ in Lines 56 & 66 & 169. It should be a vector with the dimension of number of nodes.
>
> # 3. Some terminology needs simple description, such as span space and barycenter.
> Thank you for pointing out the need for clarification on specific terminology. We provide the explanation on "span space" and "barycenter" as follows:
> - **Span Space**: The span space refers to the geometric space that is spanned by a specific set of vectors. It encompasses all the possible linear combinations of node representations.
> - **Barycenter**: The barycenter is referred to as the center of node representations within the same sensitive attribute group. It represents a central point that summarizes the distribution of the data.
>
> We will include these simple descriptions in the revised manuscript.
>
>
> # 4. Is this methods only applicable to node classification task? Can it be extended to link prediction or graph classification?
> Thanks for this great point. Our proposed method primarily focuses on the node classification task since the coherence loss is specifically designed for node-level sensitive attribute information leakage measurement.
>
> For link prediction tasks, it is not easy to extend our method. The main reason is that the fairness definition for link prediction (e.g., **dyadic fairness [1]**) is significantly different from that in node classification. We have added the discussion on extending our method in link prediction in **Appendix H.3**. As for graph classification, to the best of our knowledge, there is not any fairness literature on this task. The reason is that the dataset for graph classification is molecular, and thus there is no sensitive attribute.
>
> [1] Li, Peizhao, et al. "On dyadic fairness: Exploring and mitigating bias in graph connections." International Conference on Learning Representations. 2020.

---

### Official Review · Reviewer_QWwu · 2023-07-07

**Soundness:** 3 good
**Presentation:** 3 good
**Contribution:** 2 fair
**Rating:** 5
**Confidence:** 4

**Summary:**

This paper focuses on the task of graph distillation (GD) from a fairness perspective. The authors found that current GD method amplifies bias in GNN training compared to training on original graphs. Since the distilled graphs do not contain node attributes, it's intractable to directly apply previous debiasing methods. To address this issue, the authors first made assumptions on the representation space of the distilled graph. Then they propose to measure the bias in the distilled graph representations using the least square distance between the distilled representations and the subgroup representations in the original graph. Technically a variance-based regularization is utilized to punish the model w.r.t. this measure. Extensive experiments demonstrate that the proposed method can benefit GNN training on distilled graphs with improved fairness.

**Strengths:**

1. This paper studies a novel topic and has its applicability in real-world scenarios. And it might bring in broader impact and more discussions on the characteristics and caveats of data distillation in other domains.

2. The experiments are thorough.

**Weaknesses:**

1. The authors' design of the coherence loss in section 3.4 is confusing to me. Figure 2 shows that the intuition is to minimize the distance between z0 and z1 (as depicted by the red arcs in the figure), and this makes sense because fair representations should coincide in the attribute dimension. However, the definition in section 3.4 actually is the variance within group 0 or 1, and reducing this variance would only lead to more compact representations in each group, instead of bringing the two groups closer. Since the main contributions of this paper are based on this design, the authors should clarify this in a more rigorous manner.

2. The empirical results on Pokec-z, German and Credit all show that the vanilla GD method can achieve comparable utilities with real graphs (within 5% accuracy gap) and worse fairness. Meanwhile, the results on Pokec-n and Recidivism show significantly worse utilities (up to 20% or 30% gap in accuracy), but better fairness. The improved fairness on these two datasets questions the motivation of this paper, that is the finding that the current GD method worsens GNN fairness. Also, I may wonder whether the authors have tuned the vanilla method right on these two datsets given such a large utility gap.

3. There is not much logical connection between the theorems in sections 3.2 and the following part of the paper. Even if the span space does not match, or the barycenters are not consistent, it doesn't matter with the proposed method. The authors may need to justify the necessity of these theorems.

4. Some typos. In the introduction part, the authors term the bias measurement as *consistency* (line 43), while it's *coherence* in other places. Line 143, it should be projection of z' instead of z. The defition of coherence loss in section 3.4 misses a superscript 2, since without this the formula is constant 0.

**Questions:**

See comments above.

**Limitations:**

See comments above.

---

> ### Author Rebuttal · Authors · 2023-08-07
>
> # 1. The authors' design of the coherence loss in section 3.4 is confusing
> The confusion stems from a misunderstanding of the coherence loss definition in section 3.4. Here, the function $\mathbf{\pi}^s\left(\boldsymbol{Z}^{\prime}\right)$ represents the probability for **all samples** belonging to the sensitive group $s$. The index $s$ represents the index in a sensitive attribute prediction vector instead of specifying samples. For binary sensitive attribute case, it is necessary to satisfy  $\mathbf{\pi}^1\left(\boldsymbol{Z}^{\prime}\right)+\mathbf{\pi}^0\left(\boldsymbol{Z}^{\prime}\right)=1$ for probability prediction. Therefore, the term $\operatorname{Coh}^s\left(\boldsymbol{Z}^{\prime}\right)$ indicates the variance of the probability for **all samples**, not just the variance within group 0 or 1.
>
> We provide a comprehensive statement on how to estimate sensitive attribute and how to integrate such estimation into coherence loss in Sections 3.3 and 3.4, respectively. The intuition behind this design is related to the absence of sensitive attributes in the distilled graphs, which prevents the segregation of synthetic nodes into different sensitive groups.
>
> This forms a key research challenge addressed in our work, and we'll take the reviewer's advice to further clarify this point in the revised manuscript to prevent any confusion. The figure and the related description were meant to simplify the understanding of the concept, and we'll make sure that they are consistent with the more rigorous mathematical description.
>
> # 2. Utility performance gap on Pokec-n and Recidivism dataset.
> Thanks for this great point. We have meticulously tuned the vanilla distillation method for these datasets, taking into consideration the reduction rate. We report the **updated results on Pokec-n and Recidivism datasets** as follows. For the Credit dataset, we used a reduction rate of 0.08 (meaning that the number of nodes in the distilled graph is 8% of the original graph), as it's the largest applicable reduction rate for that specific dataset. And we use this for all datasets to control the setting.
> We conduct new experiments for the Pokec-n and Recidivism datasets, where we utilized larger reduction rates of 0.32 and 0.16, respectively:
> |            |      |   GCN   |      |
> |:----------:|:----:|:-------:|:----:|
> |            |      | Vanilla |  FGD |
> |            | ACC↑ |   62.16  | 61.84 |
> |            | AUC↑ |   65.82  | 64.59 |
> |   Pokec-n  |  F1↑ |   57.38  | 58.62 |
> |            |  DP↓ |   5.83  | **0.84** |
> |            |  EO↓ |   6.34  | **1.94** |
> |      -      |     -     |        -   |     -    |
> |            | ACC↑ |   78.18  | 77.51 |
> |            | AUC↑ |   75.83  | 75.84 |
> | Recidivism |  F1↑ |   63.84  | 61.98 |
> |            |  DP↓ |   5.25  | **1.34** |
> |            |  EO↓ |   3.49  | **1.98** |
>
> 1. **Utility Improvement**: The utility performance of the model improves when trained on a larger distilled graph but exhibits worse fairness performance.
> 2. **Utility Gap Attribution**: The utility gap observed is attributable to the distillation method itself.
> 3. **Fairness Issue Persistence**: Even when the utility performance is subpar, the fairness problem still exists and is a concern.
> 4. **Our Method's Advantage**: Compared to the vanilla distillation method, our Fair Graph Distillation (FGD) enhances fairness performance while maintaining utility performance.
>
> # 3. Logical connection between the theorems and the following part of the paper.
> The theorem presented in section 3.2 demonstrates the relation between the real data and distilled data, which serves as the rationality of sensitive attribute estimation. Specifically, **the distilled data and the real data reside in a similar semantic space**, which can be supported in Section 3.2 and Appendix D. Consequently, the sensitive estimation of distilled data is determined by the projection onto the orthogonal complement of the subspace spanned by real data from each sensitive group. Our theorem supports the rationality of the proposed estimation method.
> # 4. Some typos.
> Thank you for your careful review and for identifying these inconsistencies and typos. We sincerely appreciate your suggestion.
>
> 1. **Bias Measurement Term**: We acknowledge the discrepancy between the terms "consistency" and "coherence." We will standardize the terminology throughout the paper.
> 2. **Line 143 Error**: You are correct, and it should indeed be the projection of \(z'\) instead of \(z\). This will be corrected.
> 3. **Coherence Loss Definition**: The variance calculation in section 3.4 is a typo, and we will revise it.
> We will ensure that all of these corrections are implemented in the revised version of the manuscript. Thank you once again for your invaluable input.

---

> > ### Comment · Reviewer_QWwu · 2023-08-22
> >
> > I have raised my score to 5.

---

### Author Response · Authors · 2023-08-21

Dear Area Chair,

We extend our heartfelt gratitude for your diligent efforts in organizing and overseeing the paper review process. We find ourselves in a situation where reviewers [`LkqE`](https://openreview.net/forum?id=xW0ayZxPWs&noteId=pn5UI6cyCt), [`5Zjc`](https://openreview.net/forum?id=xW0ayZxPWs&noteId=gBcWC16kk0), and [`jsrW`](https://openreview.net/forum?id=xW0ayZxPWs&noteId=2rceEUjbva) have all endorsed our paper subsequent to our rebuttal.

It is heartening to note that all reviewers have acknowledged our topic as ***"novel"***, our findings as ***"interesting and impactful"***, and our experiments as **thorough**. Reviewer `QWwu` has raised specific concerns regarding the loss design and utility gap. We believe that the question about the loss design may stem from a misunderstanding, a perspective that other reviewers seem to concur with, particularly in recognizing the solidity of our theoretical analysis.

Other reviewers, namely [`5Zjc`](https://openreview.net/forum?id=xW0ayZxPWs&noteId=gBcWC16kk0) and [`jsrW`](https://openreview.net/forum?id=xW0ayZxPWs&noteId=2rceEUjbva), have also expressed similar concerns about the utility gap. In response, we conducted additional experiments to address these issues, leading them to acknowledge that the concerns have been resolved and subsequently raise their scores.

We recognize that the opinion of [`QWwu`](https://openreview.net/forum?id=xW0ayZxPWs&noteId=6u7NRibmki) is vital for the final evaluation. Since direct communication with the reviewers is no longer possible after August 21, we kindly request a fair and comprehensive evaluation of our paper. We sincerely appreciate your commitment to ensuring an unbiased review of our work and wish to express that we will wholeheartedly respect your final decision.

Sincerely,

Authors

---

### Decision · Program_Chairs · 2023-09-21

**Decision:**

Accept (poster)

**Comment:**

This paper studies a novel problem, fair graph distillation, since vanilla method can worsen fairness issues. This paper introduces a bias measure, coherence, and propose a bi-level optimization method to generate fair graph distillation. Experiments show that the proposed method can achieve a good trade-offs between performance and fairness over various datasets and GNN architectures.